# Experimental Study on Forged TC4 Titanium Alloy Fatigue Properties under Three-Point Bending and Life Prediction

**DOI:** 10.3390/ma14185329

**Published:** 2021-09-15

**Authors:** Bohan Wang, Li Cheng, Dongchun Li

**Affiliations:** Aeronautics Engineering College, Air Force Engineering University, Xi’an 710038, China; cheng_qiaochu@foxmail.com (L.C.); lidongdong313@163.com (D.L.)

**Keywords:** forging, TC4 titanium alloy, three-point bending, very high cycle fatigue, microstructure, life prediction

## Abstract

Ultrasonic fatigue tests of TC4 titanium alloy equiaxed I, II and bimodal I, II obtained by different forging processes were carried out in the range from 10^5^ to 10^9^ cycles using 20 kHz three-point bending. The results showed that the S-N curves had different shapes, there was no traditional fatigue limit, and the bimodal I had the best comprehensive fatigue performance. The fracture morphology was analyzed by SEM, and it was found that the fatigue cracks originated from the surface or subsurface facets, showing a transgranular quasi-cleavage fracture mechanism. EDS analysis showed that the facets were formed by the cleavage of primary α grains, and the fatigue cracks originated from the primary α grain preferred textures, rather than the primary α grain clusters. From the microstructure perspective, the reasons for better equiaxed high-cycle-fatigue properties and better bimodal ultra-high-cycle-fatigue properties were analyzed. The bimodal I fatigue life prediction based on energy was also completed, and the prediction curve was basically consistent with the experimental data.

## 1. Introduction

With the continuous improvement of aviation equipment reliability and life index, the ultra-high-cycle-fatigue (10^7^~10^12^ cycles) problem has attracted increasingly more attention, and has become the focus of fatigue research [1,2,3,4,5]. Ultra-high-cycle fatigue belongs to micro-scale fatigue, which differs from traditional high-cycle fatigue (10^5^~10^7^ cycles) not only in the life length, but also in the more complex crack initiation and initial propagation mechanism, and also in the great difference due to different materials [6,7,8]. Titanium alloys have become the most widely used metal materials in the modern aviation industry because of their high specific strength, low density, excellent corrosion resistance, and good heat resistance, and the ultra-high-cycle fatigue problems are also the most prominent. Among them, TC4 titanium alloy is widely used in the manufacture of aero-engine compressor blades. The engine rotating speed is as high as tens of thousands of revolutions per minute, and it is very easy for rotating parts such as blades and discs to cause high-frequency flutter under the internal flow disturbance and rotor imbalance action, and ultra-high-cycle-fatigue fracture problems are easy to occur when accumulating for a long time [9,10]. This shows that the safety of aero-engine compressor design cannot be ensured according to the traditional 10^7^ fatigue limit design theory.

Due to the observation method limitations and the theoretical analysis complexity, ultra-high-cycle-fatigue studies basically adopt phenomenological analysis methods based on experimental results, including macroscopic analysis of the S-N curve and Goodman diagram based on experimental results, and microscopic analysis of crack initiation and initial propagation based on fracture morphology. The proportion of crack initiation life is more than 95% in the high-cycle-fatigue and ultra-high-cycle-fatigue regimes [11,12]; therefore, the crack initiation mechanism and its relationship with fatigue life are very important. A large number of studies have shown that [13,14,15] the remarkable crack initiation zone sign for titanium alloys is the smooth facet morphology, especially for failure on the subsurface. The reasons for the facet formation involve three different mechanisms: cleavage [16,17], slip [18,19], and twins [20], which are highly controversial at present. Researchers have also made in-depth analyses based on the test results, constructed many ultra-high-cycle-fatigue fracture analysis models of titanium alloys, and predicted the life [21,22]. However, most studies have focused on the axial tension-compression loading mode, which is not consistent with the bending vibration mode of aero-engine blades, so whether these analytical models are suitable for three-point bending loading remains to be further studied.

Forging is the most widely used method for the plastic forming of titanium alloys, and it is also a key link in the blade manufacturing process. The forging performance is mainly determined by the forging process. Once a bad microstructure is formed in the forging process, the subsequent heat treatment is difficult to improve. In addition, titanium alloys are very sensitive to the forging process, in which temperature determines the solid phase transformation behavior of titanium alloys, and the deformation degree and deformation rate also affect the morphology, size, proportion, and distribution of α and β phases [23,24]. Tanaka et al. [25] studied the quantitative relationship between the Ti-17 titanium alloy fatigue properties and microstructure at different forging temperatures. It was found that the fatigue strength is closely related to the microstructure factors such as the volume fraction of the equiaxed α phase, and it is one of the crack initiation sites during low-temperature solid solution aging. The strength difference between the acicular α phase and fine α + β phase is the main reason for crack initiation after high-temperature solid solution aging. Nikitin et al. [26,27] studied the ultra-high-cycle-fatigue crack initiation mechanism of forged VT3-1 titanium alloy under tension and torsion. It was found that the traditional fatigue limit does not exist and the crack initiation position changes from surface to subsurface. When the stress ratio is *R* = −1, the subsurface cracks initiate at the strong defects, “macro region” boundaries, quasi-facets, and facets, while when the stress ratio is *R* = 0.1, the cracks mainly initiate at the “macro region” boundaries. Sinha et al. [28] studied the relationship between the fatigue life scatter and fracture mechanism of the forged Ti-6242Si alloy, and clarified the microcrack initiation and propagation process by the quantitative fatigue fracture characterization. Combined with the crystallographic characteristics analysis of the crack initiation facet and adjacent facet, it was concluded that the difference in cracking degrees along the primary α grains base plane leads to great specimen fatigue life variability.

In this paper, the forged TC4 titanium alloy three-point bending fatigue properties were studied. Equiaxed I, II and bimodal I, II were obtained by four forging processes. In addition, 20 kHz room-temperature ultrasonic fatigue tests were carried out, and the corresponding S-N curves and fatigue fractures were obtained. The crack initiation mechanism was revealed by fracture SEM + EDS analysis, and the microstructure effect on fatigue properties was analyzed by the changing trend of S-N curves. The bimodal I life prediction based on the energy method was also carried out. The work performed was helpful to determine the forging process with the best anti-fatigue ability of aero-engine blade materials.

## 2. Materials and Methods

### 2.1. Materials

The raw material was aviation-grade TC4 bar with a diameter of 28 mm, and its chemical composition is shown in Table 1. The TC4 titanium alloy phase transition temperature measured by the continuous temperature-increasing metallographic method was 998 °C. On this basis, four process combinations of two temperatures (950 °C and 985 °C) and two deformations (39.3% and 69.6%) were selected for plate die forging. The single piece blanking size was 150 mm, and the bars were heated to the specified temperature for 15~20 min and then flattened along the radial direction. After forging, the bars were annealed at 720 °C × 1 h + AC.

Figure 1 shows the microstructures obtained by four forging processes. It was found that the forging temperature has a great influence on the microstructure morphology, while the effect of deformation is relatively small. The equiaxed primary α phase can be clearly observed, and the remaining phases are the β transformation microstructure, including the fine acicular secondary α phase and residual β phase. No obvious defects such as crack, inclusion, segregation, and folding were found. The primary α content, grain size, and standard deviation were measured, and the results are shown in Table 2. The equiaxed I, II and bimodal I, II were obtained.

The room-temperature tensile properties of four kinds of forgings were obtained by a WDW-300 universal testing machine, and the test results were averaged for 3 times. The stress–strain curves were drawn according to the relationship between tension and displacement during loading, as shown in Figure 2. Finally, the tensile strength *σ*_R_, yield strength *σ*_0.2_, elongation *A*, section shrinkage *Z*, elastic modulus *E*, and Poisson’s ratio *v* were obtained, as shown in Table 3.

### 2.2. Methods

Finite element modal analysis was used to carry out the specimen design, the linear perturbation analysis step was selected after the three-dimensional model was introduced, the first 10 free vibration modes were extracted, the stress and displacement were set as output variables, and the structured hexahedral grid was divided. Finally, the modal analysis results of the 7th-order three-point bending vibration were obtained, as shown in Figure 3. The modal frequency was 19,954 Hz, which is close to the design frequency of 20 kHz. The middle area of the specimen bottom bore the maximum tensile stress, which is the expected fracture position, and the two displacement standing points were symmetrical with respect to the middle section, which are the fulcrum positions. By modifying the specimen length to optimize the resonant frequency, it was determined that the four forged specimen lengths *L* were 31.5, 31.6, 31.8, and 31.7 mm respectively, as shown in Figure 4.

The tests were carried out by using the HC-DF2020GD-K2 multi-function ultrasonic fatigue testing machine (HC SONIC, Hangzhou, China). The loading frequency was 18.5~20.5 kHz, the static loading range was 0~10 kN, and the force sensor control accuracy was 0.5%. The horn output amplitude was 10~150 μm, and the amplitude control precision was 1 μm. The three-point bending loading process is shown in Figure 5. The middle position of the specimen bottom bore both bending stress *σ*_m_ and vibration stress *σ*_a_, and the stress ratio *R* = (*σ*_m_ − *σ*_a_)/(*σ*_m_ + *σ*_a_). The relationship between *σ*_m_ and static pressure *P* was calibrated by a strain gauge. *σ*_a_ is proportional to the horn output amplitude *A*_0_, and the ratio was obtained from the stress–displacement nephogram in Figure 3. The fatigue test started from the high-stress area, and the stress amplitude was gradually reduced by 15 MPa until the cycle reached 1 × 10^9^, and each stress level was subjected to 1 to 2 samples. The output amplitudes under different ultrasonic powers were measured by an LV-S01 laser vibrometer (SOPTOP, Ningbo, China) and used to calibrate the control software. In order to restrain the temperature rise caused by high frequency and ensure the stability of the fatigue test, the specimens were cooled by water mist.

The Apreo S SEM (FEI, Hillsboro, OR, USA) was used to observe the fracture morphology, and through the energy spectrum module installed on it, EDS analysis of the ultra-high-cycle fatigue crack origin zones of the four microstructures was carried out.

## 3. Results

### 3.1. S-N Curves

As the number of loading cycles increases, the crack initiation changes from surface to subsurface, which shows that the difference between ultra-high-cycle-fatigue and high-cycle-fatigue lies not only in lower stress and longer life, but also in the fatigue failure mechanism. The S-N curves of the four microstructures include the single-step decline type, double-step decline type, and straight-step decline type, and there is no traditional fatigue limit. The equiaxed high-cycle-fatigue properties are obviously better than those of bimodal microstructures, while the equiaxed I ultra-high-cycle-fatigue property is worst. The equiaxed II ultra-high-cycle-fatigue property is similar to that of bimodal I, but the fatigue life scatter is larger, so it is considered that the bimodal ultra-high-cycle-fatigue properties are better than those of equiaxed microstructures. From the analysis in Figure 6, it seems that the high-cycle-fatigue and ultra-high-cycle-fatigue performances cannot be obtained at the same time. In the range of 10^5^ to 10^9^ cycles, bimodal I has better comprehensive fatigue performance, and 985 °C with a 39.3% deformation degree is determined to be the best forging process.

### 3.2. Fatigue Fracture Morphology

Figure 7a–d show the high-cycle-fatigue crack origin zone SEM morphology of different forged TC4 titanium alloys at low magnification. It is observed that the river-like crack propagation patterns converge on the specimen surface, in which the bimodal I has a bright and continuous main crack, and the propagation direction is almost vertical. There are no obvious inclusion traces in the origin zones, which is significantly different from the fisheye morphology in high-strength steel. Figure 7e–h show the high-cycle-fatigue crack origin zone SEM morphology at high magnification, and many smooth facet features are observed, especially on the surface, which are considered to be the crack origins. There are a large number of cleavage steps and microplastic tear (MPT) morphologies around the facets. Overall, there is no obvious difference in the crack origin zone SEM morphology for the four kinds of microstructures, and all of them are mainly transgranular brittle fracture. Combined with the crack propagation path, it is determined that the crack first initiates from the surface facet, initially propagates with the cleavage steps and MPT characteristics, and then converges with the subsurface facet cracks to form the main crack, which finally leads to fatigue fracture, which is shown as the transgranular quasi-cleavage fracture mechanism.

Figure 8a–d show the ultra-high-cycle-fatigue crack origin zone SEM morphology of different forged TC4 titanium alloys at low magnification. The crack still propagates in a river shape, but converges on the specimen subsurface, in which the equiaxed II main crack patterns are the most obvious, and the propagation direction is oblique 45°. Figure 8e–h show the ultra-high-cycle-fatigue crack origin zone SEM morphology at high magnification, and the origin zone center is the facet cluster morphology, not the inclusion characteristics. There are a large number of cleavage steps and MPT features around the facets. The analysis shows that the near small facets converge into large facets through the cleavage steps, while the distant large facets are connected by MPT characteristics, and the whole is still dominated by transgranular brittle fracture. The distance between the subsurface facet and the surface of the four kinds of microstructures is about 48, 39, 45, and 43 μm. Combined with the crack propagation path, it is concluded that the crack first originates from the subsurface facets, and then the faceted cracks with different heights and orientations connect and propagate each other through the cleavage steps and MPT features, and finally merge with the main crack to lead to fatigue fracture, which is still the transgranular quasi-cleavage fracture mechanism.

The fatigue crack initial propagation depth is about between 71 μm~273 μm, and then it will enter the slow and stable propagation stage. The crack propagation region is relatively flat, and the slow propagation zone shows a mixed shape of dense shear tear edges and secondary cracks. Due to the low crack propagation rate, the fatigue band is not obvious, but there are a large number of secondary cracks, indicating that the crack direction changes after it encounters the second phase or grains with different orientations in the crack propagation process. The stable propagation region is composed of many small fault blocks with different sizes and heights, on which there exist thin and short fatigue striations, as shown in Figure 9. A series of basically parallel and slightly curved fatigue striations are clearly visible. Each fatigue striation represents the crack tip position under the cycle, and the number of fatigue striations is roughly equal to the number of cycles and perpendicular to the local crack propagation direction. The high-cycle-fatigue striation spacings of the four microstructures are 0.097, 0.139, 0.219, and 0.140 μm, respectively, and the ultra-high-cycle-fatigue striation spacings are 0.095, 0.087, 0.118, and 0.086 μm, respectively. For the same microstructure, the distance between the fatigue striation observation position and the origin zone is basically the same, so the fatigue striation spacing in the local microzone basically reflects the crack propagation rate. The ultra-high-cycle-fatigue striation spacing is smaller than that of high-cycle-fatigue, so the crack propagation life is longer.

## 4. Discussion

### 4.1. Analysis of Crack Initiation Mechanism

For TC4 titanium alloy, Al and V are the stable elements of the α and β phase, respectively, that is, the Al content in the α phase should be higher than that of the matrix, and the V content should be lower. EDS analysis was carried out on the facets in the ultra-high-cycle-fatigue crack origin zones of the four kinds of microstructures, and the results are shown in Figure 10. Among the 24 randomly selected facets, the V content is lower than the matrix level, and the Al content in most facets is higher than the matrix level, while the Al content in a few facets is lower, and the missing part is replaced by the Ti element. Considering that the facets are about tens of microns, slightly smaller than the primary α grain size, it is concluded that the facets in the crack origin zone are formed by the cleavage of primary α grains. The formation of these primary α grains is mainly attributed to the Al element-promoting effect, while a few of them hardly need the Al element assistance. Combined with SEM and EDS analysis, it is concluded that the crack initiation mechanism is the cleavage of primary α grains on the surface or subsurface, corresponding to high-cycle-fatigue and ultra-high-cycle-fatigue, respectively, and the fatigue failure mechanism is mainly transgranular quasi-cleavage brittle fracture.

The facets in the crack origin zone appear in the form of clusters, rather than the single facet. Chandran [29] believed that the spatial α grain agglomeration leads to the formation of fatigue cracks in Ti-10V-2Fe-3Al titanium alloy. In order to investigate whether the TC4 titanium alloy fatigue cracks originate from a single facet or facet clusters, four kinds of microstructure ultra-high-cycle-fatigue origin zone facet clusters were analyzed by EDS. If the fatigue cracks originate in facet clusters, the Al content should be significantly higher than the matrix level, and the V content should be lower. As shown in Figure 11, the three element contents are very close to the matrix level, so it is concluded that facet clusters are not the cause of fatigue crack formation. Kun et al. [30] obtained a similar conclusion by analyzing the Ti-8Al-1Mo-1V titanium alloy inverse pole diagram along the loading direction. It was found that the grain orientation is random on the whole, but the adjacent grain orientation is basically the same in the local microzone, indicating that the fatigue cracks originate from the preferred texture of primary α grains rather than clusters. To sum up, the fatigue crack initiation mechanism is the cleavage of primary α grains with specific spatial orientation and crystal orientation for the four kinds of microstructures.

### 4.2. Analysis of Microstructure Influence

The titanium alloy fatigue properties are mainly affected by the microstructure such as primary α grains and are closely related to the forging process. On the one hand, the increased forging temperature will strengthen the primary α phase diffusion behavior, swallow the surrounding fine α phases, and cause the primary α grains to grow. On the other hand, the increased forging temperature will promote the α→β phase transformation behavior, resulting in the decrease in primary α grain size and content. The two mechanisms compete with each other under the deformation degree influence, resulting in changes in the primary α content and grain size, as shown in Table 2.

As the α phase is more brittle than the β phase is, it is easier to form dislocation accumulation at the α-β phase boundaries or primary α grain boundaries under fatigue loading. The larger primary α grain boundaries increase the dislocation slip length and aggravate the stress concentration at the tip of dislocation accumulation, thus promoting the primary α grain cleavage failure [31]. Stanzl et al. [32] believed that the accumulated irreversible slip in the process of low stress fatigue leads to the α grain cleavage fractures in the slip surface with a high Schmidt factor, and the subsequent process of faceted crack nucleation and propagation form a rough origin zone. Chai et al. [33] studied the subsurface defect-free crack initiation mechanism and obtained a similar conclusion: the primary α phase is the weak crack initiation part, and the longer primary α grain boundaries play the role of internal notches.

The analysis shows that the fatigue life is related to the microstructure influence on the high-cycle-fatigue and ultra-high-cycle-fatigue mechanism. The traditional high-cycle-fatigue belongs to macro-scale fatigue, the material composition can be approximately regarded as a uniform distribution, the fatigue failure is mainly controlled by surface stress, and the microstructure influence is small. At the same stress level, the primary α phase content in the equiaxed microstructure is higher than that in the bimodal microstructure, which can tolerate more slip deformation and enhance the resistance to crack initiation, so the high-cycle-fatigue performance is better. In contrast, the ultra-high-cycle-fatigue belongs to micro-scale fatigue, material composition is no longer uniformly distributed, fatigue failure is controlled by both surface stress and internal defects, and the microstructure influence is greater. The lower primary α phase content in the bimodal microstructure means fewer defects, which reduces the number of stress concentration zones caused by dislocation accumulation at the second phase interface, makes the slip deformation more uniform, and increases the crack initiation life. In addition, the β transition tissue content in the bimodal microstructure is higher and the creep resistance is greater. As shown in Figure 1, the strip-like and fine needle-like secondary α phases are arranged longitudinally and horizontally in the β matrix, isolating the primary α grains from each other, making the crack propagation path more tortuous, blocking the facet crack propagation, converging and combining into the main crack, and increasing the crack propagation life, so that the bimodal ultra-high-cycle-fatigue properties are better.

### 4.3. Fatigue Life Prediction Based on Energy

In the linear elastic fracture mechanics, the stress intensity factor amplitude Δ*K* at the crack tip is the main crack propagation control parameter, which can be calculated by the Murakami model [34], as shown in Formula (1):(1)$$\Delta K_{f,orFC}=n\cdot \sigma_{a}{\left(\pi \sqrt{area_{f,orFC}}\right)}^{1/2}$$

For surface cracks, *n* = 0.65; for subsurface cracks, *n* = 0.5; *σ**_a_* is the stress amplitude; and *area_f,orFC_* represents the projected area of facet or facet clusters in the principal stress direction. Based on the measurement of the facet characteristics in the bimodal I crack origin zone, it is found that there is no obvious difference in most facet sizes, with an average of about 17.5 μm, while the facet cluster size increases with the fatigue life. The stress intensity factors Δ*K_f_* and Δ*K_FC_* of facets and faceted clusters are calculated according to Formula (1). As shown in Figure 12, Δ*K_f_* decreases with the fatigue life, which is much smaller than the stress intensity factor threshold value for macroscopic crack propagation (Δ*K*_th_ = 5~6 MPa·m^1/2^). The Δ*K_FC_* remains basically unchanged with the increased fatigue life, and the Δ*K_FC_* is about 4.53 MPa·m^1/2^ for surface failure and 5.49 MPa·m^1/2^ for subsurface failure, which is mainly attributed to the adverse environmental effects. Surface cracks germinate and propagate in air and water fog, while subsurface cracks do so in vacuum, so the Δ*K_FC_* is lower when surface failure occurs. To sum up, the amplitude of the stress intensity factor Δ*K_FC_* of faceted clusters is analogized to the threshold Δ*K*_th_, which controls the macroscopic stable crack propagation.

For high-cycle-fatigue and ultra-high-cycle-fatigue regimes, the crack initiation and initial propagation stages occupy the vast majority, so the fatigue life can be estimated by the energy-based crack nucleation life model, and the calculation formula is [35]:
(2)$$N_{i}=\frac{4\pi \mu^{3}h^{2}M^{3}c}{0.005d^{3}(\Delta \sigma -2Mk)\left[\pi^{2}\Delta \sigma^{2}{\left(1-v\right)}^{2}+\xi M^{2}\mu^{2}\right]}$$
where *μ* is the shear homogeneous matrix modulus, *h* is the slip band width, *d* is the primary α grain size, Δ*σ* is the stress amplitude, 2 *Mk* is defined as the fatigue strength at 10^9^ cycles, *M* is usually taken as 2, *v* is Poisson’s ratio, and *ξ* is a numerical constant. It is known from Figure 12 that the stress intensity factor amplitude of the faceted clusters Δ*K_FC_* is approximately constant, so the crack size *c* is represented by the facet cluster size *area_FC_*^1/2^. According to the S-N curve, the unknown parameters *ξ* and *h* are fitted nonlinearly, and the results are 1.02 and 8.25 × 10^−4^, respectively. Finally, the bimodal I fatigue life prediction curve is obtained, as shown in Figure 13.

The life prediction curve related to the facet cluster size is basically consistent with the fatigue data, which verifies that the fatigue failure process is mainly consumed in the facet cluster formation stage, and only a small part of the propagation zone and final rupture zone. It is also shown that the energy-based life prediction model is not only suitable for axial tension–compression loading, but also for three-point bending loading. The analysis shows that the crack origin zone shows similar facet morphology under the two loading modes, and there is no essential difference in fatigue failure mechanism, so the relevant characteristic parameters in the life prediction model are universal. As far as the current research is concerned, it is reasonable to apply the axial tension–compression fatigue analysis model to three-point bending fatigue, and the author will further analyze and demonstrate this statement in the follow-up research.

## 5. Conclusions

In this paper, forged TC4 titanium alloy ultrasonic fatigue properties under three-point bending were studied, and the main conclusions are as follows:
As the number of loading cycles increases, the crack initiation changes from surface to subsurface initiation. The S-N curves have different shapes, and there is no traditional fatigue limit. The equiaxed high-cycle-fatigue performance is better, while the bimodal ultra-high-cycle-fatigue performance is better. The best forging process combination is 985 °C with a 39.3% deformation degree.The crack origin zones in the four kinds of microstructures show the mixed morphology of facets, cleavage steps, and MPT characteristics. The crack originates from the surface or subsurface facet, showing a transgranular quasi-cleavage fracture mechanism. The ultra-high-cycle-fatigue striation spacing is smaller than that of high-cycle-fatigue for the same microstructure, which reflects a higher crack propagation life.The facet in the crack origin zone is formed by the cleavage of primary α grains. Most of these primary α grains are attributed to the Al element-promoting effect, while a few of them hardly need the Al element assistance. The fatigue cracks originate from the preferred texture of primary α grains, rather than the primary α grain clusters.High-cycle-fatigue belongs to macro-scale fatigue, the fatigue failure is mainly controlled by surface stress, and the influence of the microstructure is small. While ultra-high-cycle-fatigue belongs to micro-scale fatigue, the fatigue failure is controlled by both surface stress and internal defects, and the influence of the microstructure is great. This makes the equiaxed and bimodal microstructure have better high-cycle-fatigue and ultra-high-cycle-fatigue performance, respectively.The stress intensity factor amplitude Δ*K_FC_* of faceted clusters can be analogized to the threshold Δ*K*_th_ to control the macroscopic stable crack propagation, and the life prediction curve related to the faceted cluster size is basically consistent with the fatigue data, which shows that it is reasonable to apply the axial tension–compression fatigue analysis model to three-point bending fatigue, which is attributed to the similar fatigue failure mechanism.


## Figures and Tables

**Figure 1 materials-14-05329-f001:**
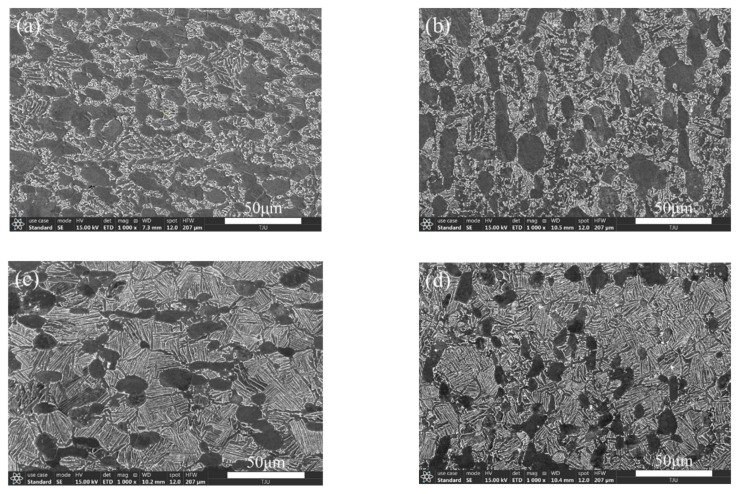
Microstructures of different forged TC4 titanium alloys. (**a**) Equiaxed I; (**b**) equiaxed II; (**c**) bimodal I; (**d**) bimodal II.

**Figure 2 materials-14-05329-f002:**
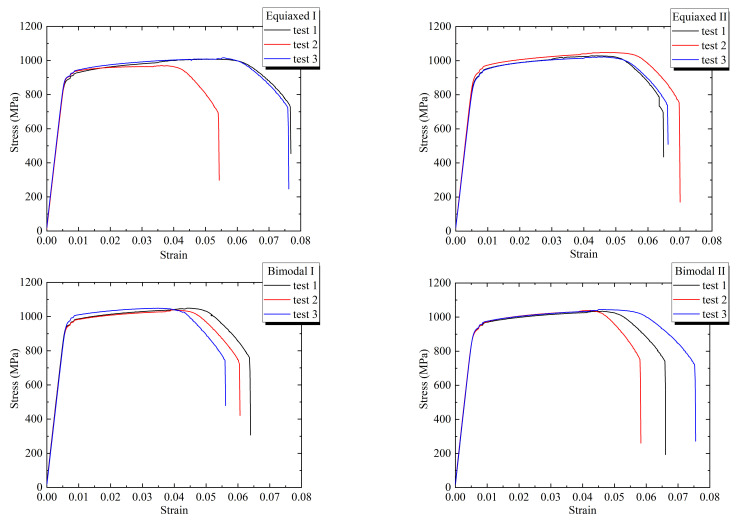
Stress–strain curves of different forged TC4 titanium alloys.

**Figure 3 materials-14-05329-f003:**
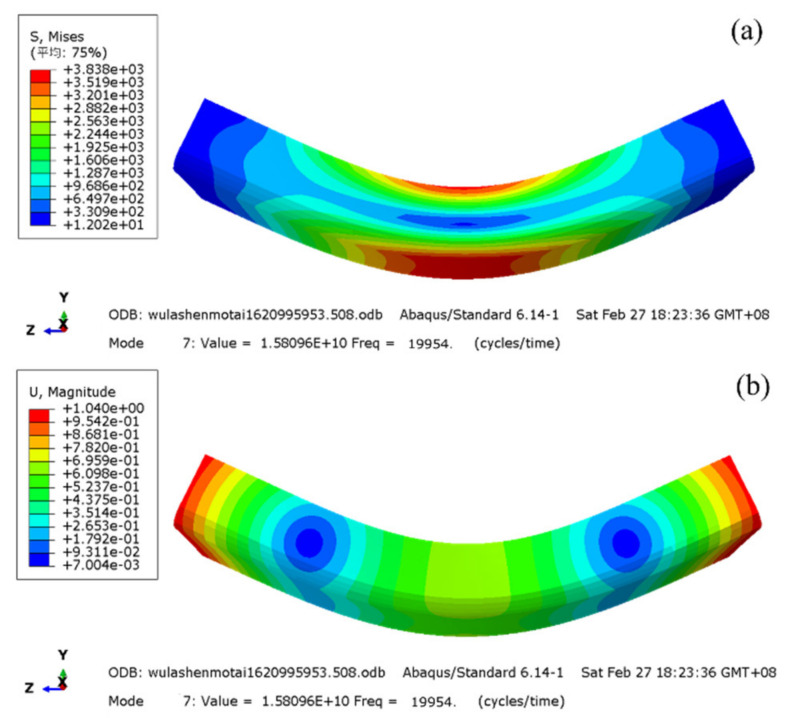
Modal analysis nephogram of equiaxial I. (**a**) Stress distribution; (**b**) displacement distribution.

**Figure 4 materials-14-05329-f004:**
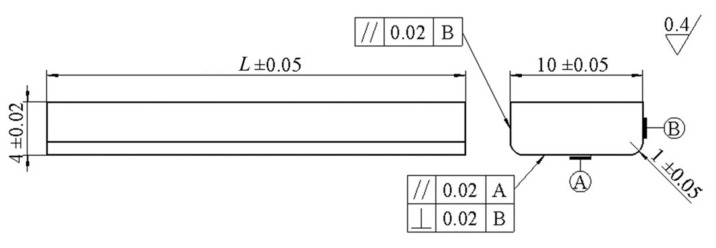
Design drawing of three-point bending specimen.

**Figure 5 materials-14-05329-f005:**
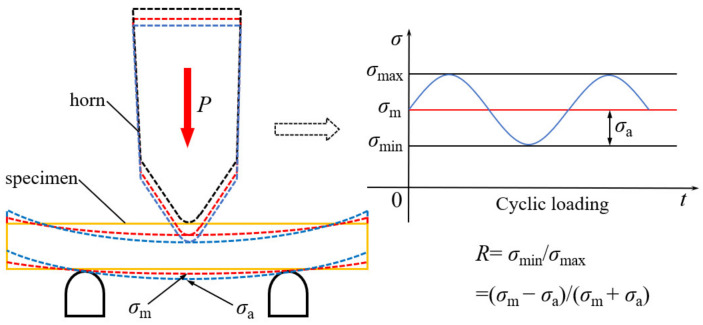
Schematic diagram of three-point bending loading process.

**Figure 6 materials-14-05329-f006:**
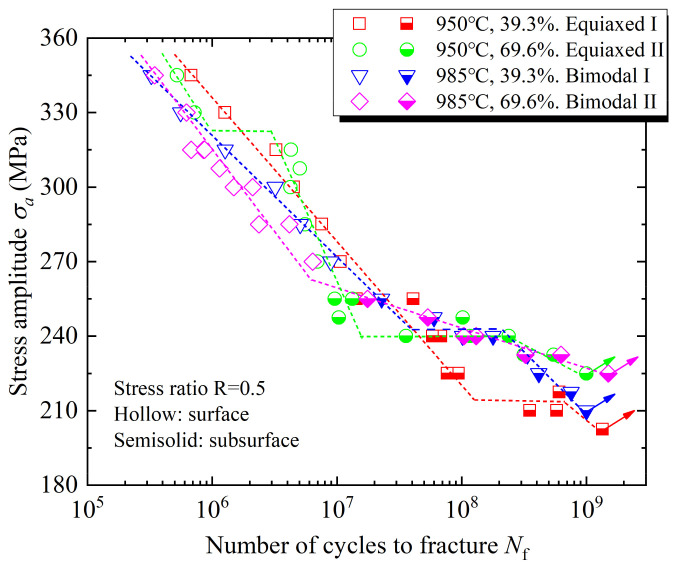
S-N curves of different forged TC4 titanium alloys.

**Figure 7 materials-14-05329-f007:**
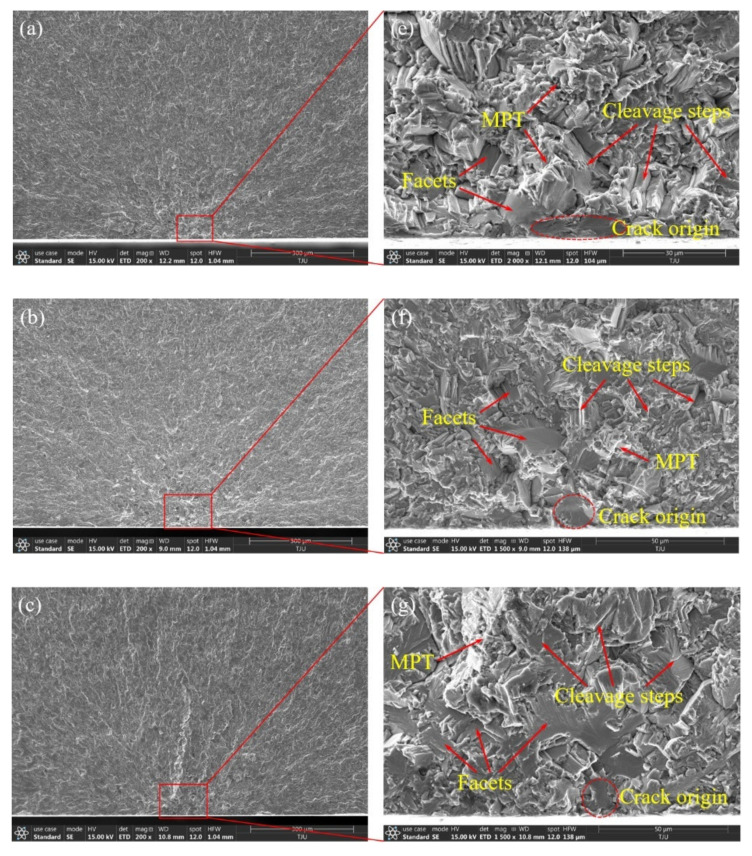
SEM morphology of high-cycle-fatigue crack origin zone. (**a**,**e**) Equiaxed I, σ_a_ = 330 MPa, *N*_f_ = 1.26 × 10^6^; (**b**,**f**) equiaxed II, σ_a_ = 300 MPa, *N*_f_ = 4.25 × 10^6^; (**c**,**g**) bimodal I, σ_a_ = 285 MPa, *N*_f_ = 5.08 × 10^6^; (**d**,**h**) bimodal II, σ_a_ = 270 MPa, *N*_f_ = 6.39 × 10^6^.

**Figure 8 materials-14-05329-f008:**
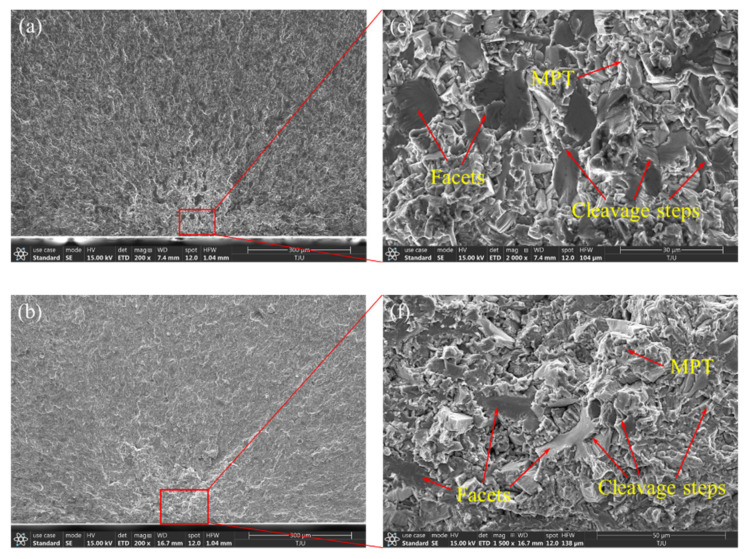
SEM morphology of ultra-high-cycle-fatigue crack origin zone. (**a**,**e**) Equiaxed I, σ_a_ = 218 MPa, *N*_f_ = 6.06 × 10^8^; (**b**,**d**) equiaxed II, σ_a_ = 233 MPa, *N*_f_ = 3.16 × 10^8^; (**c**,**f**) bimodal I, σ_a_ = 225 MPa, *N*_f_ = 4.15 × 10^8^; (**g**,**h**) bimodal II, σ_a_ = 233 MPa, *N*_f_ = 3.25 × 10^8^.

**Figure 9 materials-14-05329-f009:**
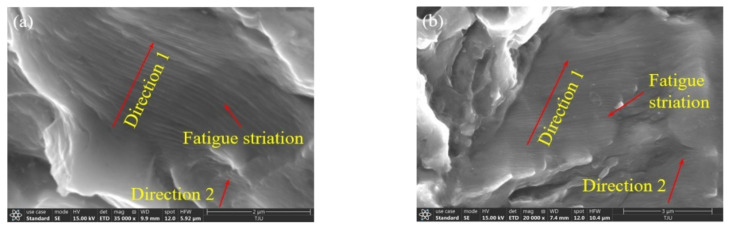
SEM morphology of fatigue crack stable propagation zone. (**a**) Equiaxed I, σ_a_ = 330 MPa, *N*_f_ = 1.26 × 10^6^; (**b**) equiaxed I, σ_a_ = 218 MPa, *N*_f_ = 6.06 × 10^8^; (**c**) equiaxed II, σ_a_ = 300 MPa, *N*_f_ = 4.25 × 10^6^; (**d**) equiaxed II, σ_a_ = 233 MPa, *N*_f_ = 3.16 × 10^8^; (**e**) bimodal I, σ_a_ = 285 MPa, *N*_f_ = 5.08 × 10^6^; (**f**) bimodal I, σ_a_ = 225 MPa, *N*_f_ = 4.15 × 10^8^; (**g**) bimodal II, σ_a_ = 270 MPa, *N*_f_ = 6.39 × 10^6^; (**h**) bimodal II, σ_a_ = 233 MPa, *N*_f_ = 3.25 × 10^8^.

**Figure 10 materials-14-05329-f010:**
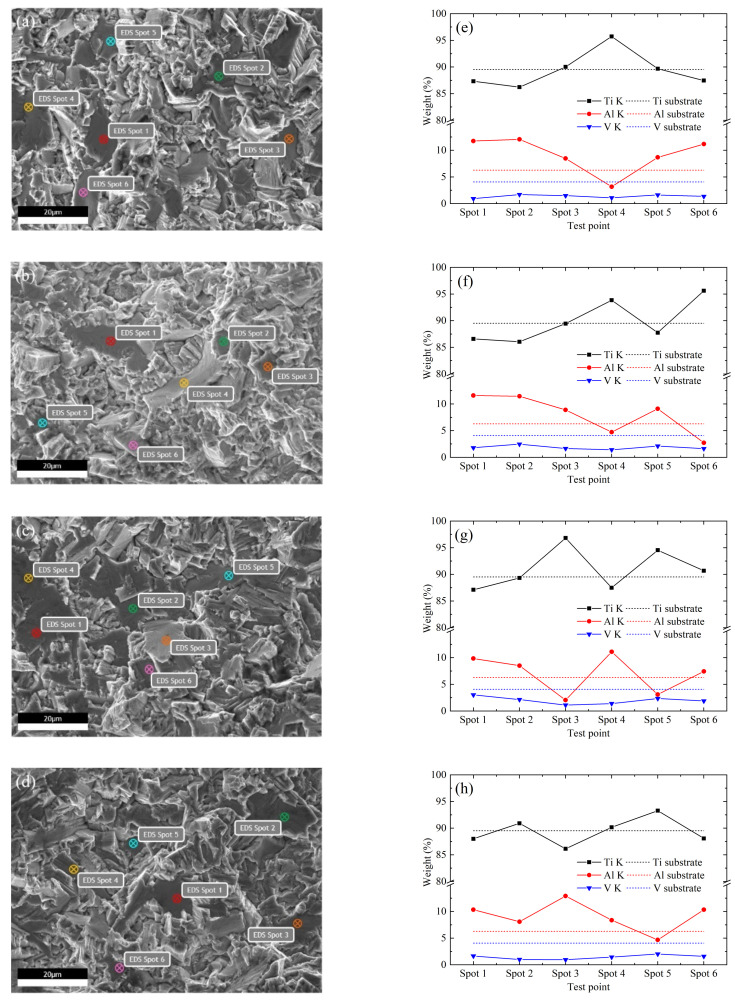
EDS analysis of facets in ultra-high-cycle-fatigue crack origin zone. (**a**,**e**) Equiaxed I, σ_a_ = 218 MPa, *N*_f_ = 6.06 × 10^8^; (**b**,**d**) equiaxed II, σ_a_ = 233 MPa, *N*_f_ = 3.16 × 10^8^; (**c**,**f**) bimodal I, σ_a_ = 225 MPa, *N*_f_ = 4.15 × 10^8^; (**g**,**h**) bimodal II, σ_a_ = 233 MPa, *N*_f_ = 3.25 × 10^8^.

**Figure 11 materials-14-05329-f011:**
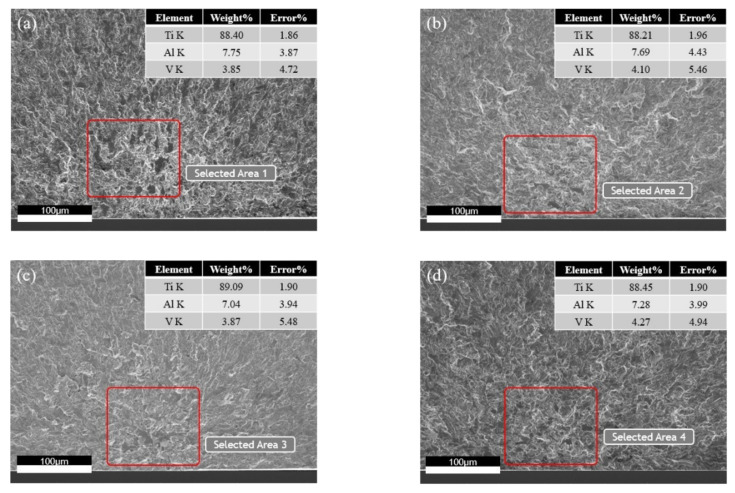
EDS analysis of facet clusters in ultra-high-cycle-fatigue crack origin zone. (**a**) Equiaxed I, σ_a_ = 218 MPa, *N*_f_ = 6.06 × 10^8^; (**b**) equiaxed II, σ_a_ = 233 MPa, *N*_f_ = 3.16 × 10^8^; (**c**) bimodal I, σ_a_ = 225 MPa, *N*_f_ = 4.15 × 10^8^; (**d**) bimodal II, σ_a_ = 233 MPa, *N*_f_ = 3.25 × 10^8^.

**Figure 12 materials-14-05329-f012:**
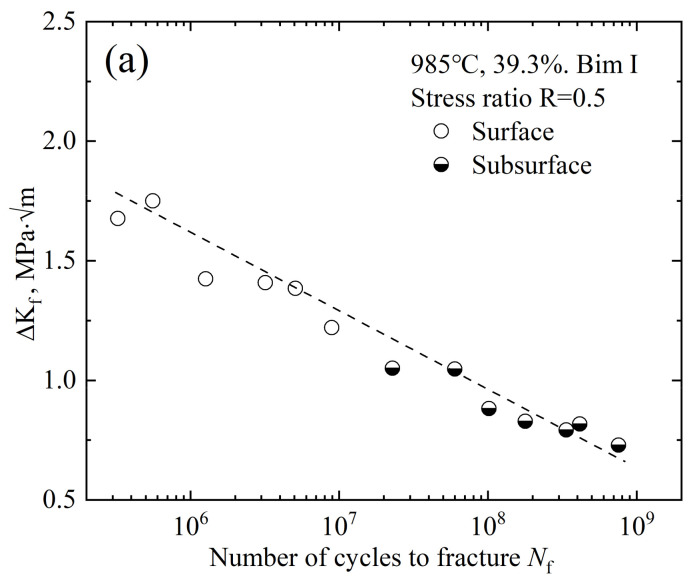
Calculation of stress intensity factor amplitude. (**a**) Δ*K_f_* and *N*_f_; (**b**) Δ*K_FC_* and *N*_f_.

**Figure 13 materials-14-05329-f013:**
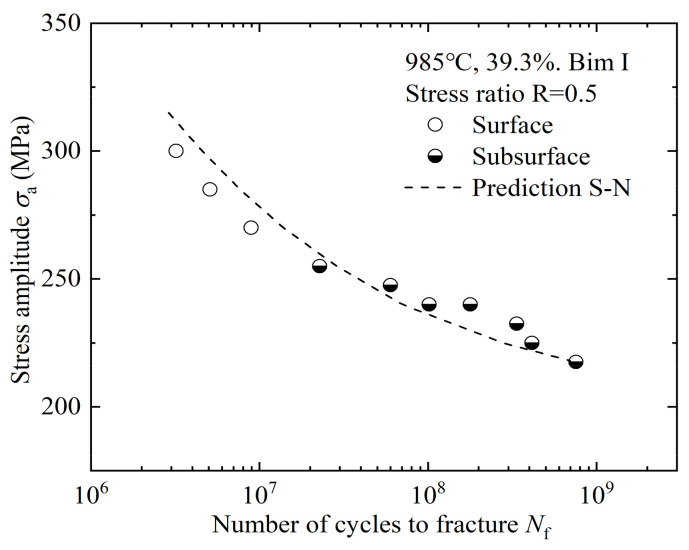
Results of the fatigue crack life prediction.

**Table 1 materials-14-05329-t001:** Chemical composition of TC4 titanium alloy (wt%).

Al	V	Fe	O	C	H	N	Ti
6.27	4.08	0.048	0.021	0.020	0.004	0.031	Bal.

**Table 2 materials-14-05329-t002:** Measurement results of primary α phase content, grain size, and standard deviation.

Forging Process	Forging Type	Content (%)	Grain Size (μm)	Standard Deviation (μm)	Microstructure
950 °C/39.3%	α + β	45.24	42.54	13.75	equiaxed I
950 °C/69.6%	α + β	48.74	34.68	16.80	equiaxed II
985 °C/39.3%	near β	27.54	23.62	19.92	bimodal I
985 °C/69.6%	near β	23.78	24.68	10.43	bimodal II

**Table 3 materials-14-05329-t003:** Room-temperature tensile properties of different forged TC4 titanium alloys.

Microstructure	σ_R_/MPa	σ_0.2_/MPa	*A*/%	*Z*/%	*E*/GPa	*v*
equiaxed I	967	898	16.2	46.8	115.85	0.317
equiaxed II	1003	951	15.8	47.6	116.83	0.329
bimodal I	986	907	14.2	51.1	120.29	0.332
bimodal II	998	924	15.2	49.5	118.91	0.324

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
