# Peer review of "Experimental Study on Forged TC4 Titanium Alloy Fatigue Properties under Three-Point Bending and Life Prediction"

_materials, 2021, doi:10.3390/ma14185329_

Round 1

Reviewer 1 Report

The paper has a good explanation about the materials and methods. Especially, section 2.1 is well-written with proper content. 

Question: in Figure 2. other figures tend to have similar SS curves from repeated tests, but in Equiaxed I, the test 2 is having more deviation as it has very low maximum stress value than other two tests such as test 1 and test 3. Please explain it.  

Question: "Line no: 149: the force sensor accuracy is 0.5%". Is it error tolerance? 

Suggestions: add SEM and EDS abbreviation. There is no abbreviation in the text.

The paper can be accepted after the minor corrections.

Thank You.

Author Response

Please see the attachment for revision instructions.

Reviewer 2 Report

Thank you for addressing most of the comments and suggestions. Thanks for renaming the term into “grain size”. Please add the standard deviation to the values in Table 2, and also to content…

Author Response

Reply:The standard deviation has been added to Table 2 and content.

Reviewer 3 Report

The paper is intersting both in findings and in applied methodology. These are the major comments of this reviewer:

  • a critical problem in such UHCF tests are the control of effective displacement/strain hence stress acting in a certain point: how authors do this measurement, experimentally?
  • what abput the temperature control, depending on strain apmplitudes?
  • another crucial issue is the damping of the system: how do authors consider it and make a control;
  • figure 6: any comparison with "traditional" fatigue data on the same alloy or alloy class should be added;
  • about the Fatigue Fracture Morphology: is there any reference/comparison in the literature?
  • being the test of 3PB type, the effect of stress gradient should be analysed/discussed.

Author Response

Please see the attachment for revision instructions.

This manuscript is a resubmission of an earlier submission. The following is a list of the peer review reports and author responses from that submission.

Round 1

Reviewer 1 Report

Your paper describes the very high cycle properties of Ti alloy. The experimental results are interesting, but the reviewer thinks lots of revisions should be made before accept, so the reviewer decided to reject this paper.

  1. You claimed that "most studies have focused on the axial tension-compression loading mode , which is not consistent with the bending vibration mode of aero-engine blades. For metal fatigues in high cycle to very high cycle region, the difference between axial and bending fatigue testings is just stress gradient  from the surface, resulting in size effect of fatigue strength; this is not substantial in view of fatigue mechanism. Thus, the reviewer does not agree your claim of the importance why you need to conduct fatigue tests by three-point bending loading.
  2. In Sec. 4.3, you claimed that the fatigue strength is governed by Kth. If so, the fatigue strength-life relation should be predicted by the fracture mechanics approach. You do not need to use the energy approach that is not widely accepted in researchers. Keeping consistency for discussion is important.
  3. Sometimes, the authors' choice of technical terms are not appropriate such as "fatigue initiation expansion mechanism" and "fatigue life dispersion". These are just examples, and the reviewer sometime felt difficulty for understanding your point when reading.
  4. Figure 6 is the main result of this paper. In Fig.6, the authors plotted zig-zag lines but there is no explanation about the lines. The reviewer can't understand why the authors plotted such zig-zag lines. It is well known that fatigue strengths show scatter as you insisted. Regression curves by using, for example, Basquin's equation, are better if plotted.

The reviewer recommend that this paper should be focused on the experimental results, and after revision, the reviewer hopes the authors re-submit your paper to this journal.

Reviewer 2 Report

This manuscript investigated the ultrasonic fatigue behavior of forged TC4 Ti alloy. There are many articles published by several authors even this author on the same materials and topics. So, it must be rejected.

Reviewer 3 Report

This experimental study investigates the fatigue properties of forged TC4 titanium alloy under 3-point bending. The results show that there is no endurance limit and the S-N curves are influenced by the microstructure. The fracture morphology is investigated by SEM and EDS analysis refers to the phases, where the cracks originate. Fatigue life predictions were done using the experimental results. The paper covers an important issue and is written well – the conclusion is based on the results. There are a few suggestions and comments – mostly on the structure of the paper: there are many results already shown in Materials and methods. The paper needs re-structuring.

Abstract

Line 1: please add cycles after the numbers given - …in the range of … cycles…

Introduction

Line 25: it would be good to refer to the term ultra-high-cycle-fatigue, which is also often used and bring it to the context of VHCF

Line 59: are instead of is…In addition, titanium alloys are very sensitive…

Materials and Methods

Line 90: please check and rephrase the first sentence – …of bars with diameters of 28mm, …

Line 92: who did determine the transformation temperature? …and how?

Line 93: please specify x% deformation, it is better to provide the deformation rate

Line 99 to 114: here results are already presented

Line 113: the average size should be presented as “grain size” (to add average to the term is not needed). More important is to present the standard deviation

Line 118 to 124: also results

Line 126: the finite element model for modal analysis is not descripted… and Figure 3 a and b show results

Line 142: please give more details on the number of samples per load

Line 155 / 156: before showing the results – there is no description of SEM-EDS method…equipment, number of samples, preparation… also: your surface is irregular shaped and you measured on facets not 90° to the beam – does this influence the results much?

Results

Line 223: please use a length provided in µm according to the grain size

Line 224: check if you made clear, what the definition for slow and stable growth stage is

Line 223: are you sure, that the number of fatigue striations are equal to the number of cycles… there might be some coarse striations, which do not correspond to a single loading cycle…

Discussion

Line 256/257: should be does not sound very clear… would it be better to say “is” instead of “should be”

Line 311: say “grain size” instead of “average size”

Line 363 and 366: Figure 12 should contain of (a) and (b)

Line 363: is it wise to put one dotted line to surface and subsurface results in Figure 12 (a) - (same comment is for Figure 13), there is such a clear difference presented in Figure 12 (b)